# The Health Status and Healthcare Utilization of Ethnic Germans in Russia

**DOI:** 10.3390/ijerph19010166

**Published:** 2021-12-24

**Authors:** Charlotte Arena, Christine Holmberg, Volker Winkler, Philipp Jaehn

**Affiliations:** 1Institute of Social Medicine and Epidemiology, Brandenburg Medical School Theodor Fontane, 14770 Brandenburg an der Havel, Germany; charlotte.arena@mhb-fontane.de (C.A.); christine.holmberg@mhb-fontane.de (C.H.); 2Faculty of Health Sciences Brandenburg, Brandenburg Medical School Theodor Fontane, 14770 Brandenburg an der Havel, Germany; 3Heidelberg Institute of Global Health, Heidelberg University Hospital, 69120 Heidelberg, Germany; volker.winkler@uni-heidelberg.de

**Keywords:** resettlers, migration, subjective health, smoking, diabetes, healthcare utilization

## Abstract

Ethnic German resettlers from the former Soviet Union are one of the largest migrant groups in Germany. In comparison with the majority of the German population, resettlers exhibit worse subjective health and utilize fewer preventive measures. However, there is little evidence on health among ethnic Germans who remained in Russia. Hence, the objective of this study was to determine the differences in subjective health, diabetes, smoking, and utilization of health check-ups between ethnic Germans and the majority population in Russia. We used data from the Russian Longitudinal Monitoring Survey II from 1994 to 2018 (general population of Russia *n* = 41,675, ethnic Germans *n* = 158). Multilevel logistic regression was used to calculate odds ratios (ORs) adjusted for age, sex, period, and place of residence. Analyses were furthermore stratified by the periods 1994–2005 and 2006–2018. Ethnic Germans in Russia rated their health less often as good compared with the Russian majority population (OR = 0.67, CI = 0.48–0.92). Furthermore, ethnic Germans were more likely to smoke after 2006 (OR = 1.91, CI = 1.09–3.37). Lower subjective health among ethnic Germans in Russia is in line with findings among minority populations in Europe. Increased odds of smoking after 2006 may indicate the deteriorating risk behavior of ethnic Germans in Russia.

## 1. Introduction

Non-communicable diseases account for a large proportion of today’s illnesses and are responsible for a major burden on the healthcare system [1]. Smoking behavior, diabetes, and the low utilization of health check-ups are related to an increased prevalence of non-communicable disease [2,3,4]. In particular, smoking behavior and diabetes represent important risk factors for the development of cardiovascular disease [5,6], which is one of the leading causes of death worldwide [7]. All-cause mortality can be predicted by subjective health, which is frequently operationalized using information about a person’s self-rated general health [8]. In recent years, there has been increasing interest in social determinants of health (SDH) as distinguished from biomedical risk factors. It has become clear that social and socioeconomic factors shape the risk of non-communicable diseases in meaningful ways [9]. Minority status, migration background and the impact of a migration process on subsequent generations represent social determinants of increasing importance. A growing body of literature indicates inequalities of both risk factors and frequency of non-communicable diseases between ethnic minorities or migrants and the respective majority populations [10,11,12]. Immigrants represent an example of an ethnic minority. An ethnic minority is a societal subgroup with unique social and cultural characteristics that differ from the majority population. Ethnic minorities are often faced with oppression, whether or not the group is a numerical minority [13]. The importance of a minority status for health can be observed when considering the example of people who have migrated to Germany. On the one hand, a lower prevalence of physical activity, lower self-assessed health, and lower participation in healthcare and prevention programs has been observed among migrant groups in Germany compared with the general population [14,15,16,17,18]. On the other hand, immigrants to Germany may have specific health resources compared with the general population. For example, a lower all-cause mortality rate was described among immigrants compared with the general population [15]. Associations, thereby, may be heterogeneous within ethnic minority populations. For example, use of the hemoccult test is lower among people born in Eastern Europe, but not among people born in Southern Europe, compared with the autochthonous German population [19]. The autochthonous population is the stationary population that has not migrated to or from the respective country in recent times.

People who have migrated to another country are exposed to countless influencing factors, both in their countries of origin and arrival (further referred to as sending and receiving countries), as well as through the migration process itself. The impact of migration is often studied in the receiving countries, although rarely in the country of origin. Completing the picture of migrant’s health by also studying the population of origin corresponds to a cross-national perspective. Such a perspective yields a more comprehensive picture of health behaviors, health risks, and resources among migrant populations by contrasting their health in the receiving countries with health in the sending population [20].

Resettlers are a relevant subgroup of people with an immigrant background in Germany. The term “resettlers” (in German, Aussiedler or Spätaussiedler) refers to ethnic Germans who were granted the right to migrate to Germany from Eastern European, Northern and Central Asian countries by a unique legal framework (“Bundesvertriebenengesetz”). Since 1990, they have immigrated almost exclusively from the former Soviet Union (FSU). Their populations of origin are Germans who moved to Russia in the 18th century after a call from Tsarina Catherine II to farm underpopulated regions [21,22]. They represented a minority group in the Russian Empire and the Soviet Union, as well as in modern successor states. Most resettlers migrated to Germany between 1990 and 1995. During this time, the Bundesvertriebenengesetz defined few restrictions for immigration from the FSU to Germany [18]. To this day, a total of about 500,000 ethnic Germans still reside in Russia, representing a largely recognized minority [23].

A cross-national perspective is especially important for resettlers because they represent a minority in their sending country. The health status of resettlers, therefore, cannot simply be compared with the health of the general populations of sending countries such as Russia, because it is unclear whether the health of ethnic Germans in Russia is comparable to the national average. Studies on health and healthcare utilization are only available for resettlers in Germany and have revealed significant differences compared with the autochthonous German population. Subjective perceptions of health and physical conditions were rated worse among resettlers compared with the German population [24]. Furthermore, lower participation in early detection hemoccult screening, but higher participation in mammography screening was observed [19,24,25]. However, resettlers exhibit lower all-cause mortality and a lower mortality from cardiovascular disease compared with the general population [15]. Research on the health of ethnic Germans in Russia is limited to one study that compared cancer incidence between ethnic Germans and the general population in the district of Tomsk, Russia [26].

These considerations imply the extension of studies of health and health behavior among resettlers from the FSU in Germany to ethnic Germans remaining in Russia; Russia is the second major country of origin of resettlers after Kazakhstan [27]. In this study, the question to be answered is whether there is a difference between ethnic Germans living in Russia and the general population of Russia in terms of their health status regarding smoking, diabetes, and subjective health, as well as their utilization of health check-ups.

## 2. Materials and Methods

### 2.1. Study Design and Population

Publicly available data of the Russia Longitudinal Monitoring Survey (RLMS) I and II, from 1994 to 2018, were used. The RLMS was originally designed by the G-7 countries to acquire objective and nationally representative data on social, health, and economic conditions in Russia. The RLMS employed a three-stage stratified clustered sampling design. First, 1850 pooled Raions (administrative-territorial districts) were created in a sampling frame. They contained 95.6% of the population and were considered primary sampling units (PSUs) [28].

Of the 98 PSUs selected, 63 were located in three metropolitan regions (Moscow City, Moscow Oblast and St. Petersburg City, Russia), whereas 35 were from the rest of Russia. Although intended to be conducted annually, due to lack of financing, the years 1997 and 1999 were omitted. Interviewers visited each selected household up to three times to obtain complete data. Any group of people sharing accommodation, income and expenses was defined as a “household”, including unmarried children up to 18 years old who were living outside the home for a short period of time. As many household members as possible above age 13 were interviewed about their health and activities. Information about younger household members was provided by their parents or guardians [28].

The response rate of the fifth round (first round of the second phase) was 87.6%. More than half of all households completed 10 rounds of the RLMS [28]. We based our analysis on the first observation of each individual within the entire period 1994–2018, and therefore conducted a cross-sectional study.

For our analysis, we used the first observation per individual from the panel data (*n* = 55,660). We excluded 13,242 observations that included legitimate missing information. The RLMS defines missing observations as legitimate if information was missing due to instructions to skip certain questions. The remaining data contained missing observations that were due to the participant’s refusal to answer. The variable ethnicity contained 585 (1.1%) missing observations; these were also excluded from the analyses. The total number of included observations was 41,833.

### 2.2. Variables

Subjective health, health check-ups visited, tobacco smoking and diabetes were the chosen outcomes. All these variables were assessed in every year of the survey, with little or no missing observations. With the question *“How would you evaluate your health?”*, the participants were queried about their subjective health. Answers to this question were provided on a 5-point Likert scale (very good, good, average, bad, very bad). For logistic regression, the five possible answers were trichotomized in the categories good (very good or good), average, and bad (bad or very bad). The following question asked about health check-ups: “*In the last three months have you seen a doctor for a medical check up, not because you were sick?*”. Moreover, participants were asked whether they currently smoked tobacco and whether they had been diagnosed with diabetes. The last three questions could only be answered as yes or no.

Self-identified German nationality was considered the exposure and was assessed by the following open question: “*What nationality do you consider yourself? I don’t necessarily have in mind the nationality in your passport.*” The reported nationalities “German” and “German-Jew” were considered as German nationality. All other nationalities, including no reported nationality, were contained in the comparison group.

A conceptual framework was established with confounders and mediators. The ascertained confounders are described below. To have an even age distribution that nevertheless represented individual stages of life in a meaningful way, age groups were coded as follows: ≤19 years, 20–39 years, 40–59 years, 60–79 years, and ≥80 years. Four categories for place of residency were available from the RLMS (regional center, big cities, small town, village). For descriptive analyses, a binary classification was used in which regional centers and big cities were defined as urban and all other categories as rural. Regarding sex, the possible answers were either male or female. For descriptive analyses, the year of survey was categorized into 4 periods of equal length (1994–1999, 2000–2005, 2006–2011, and 2012–2018).

The ascertained mediators are depicted hereafter. We selected education, employment, and marital status as mediators because a minority status can be regarded as an influencing factor for these social determinants. A study in Sweden found that first- and second-generation immigrants experienced discrimination in regard to employment, describing how they were less likely to be invited for a job interview [29]. Additionally, children from immigrant families received high education degrees less frequently compared with native children [30], and it has been shown that a lower level of educational attainment is linked to a higher prevalence of cardiovascular risk factors [10]. In addition to economic aspects, social support from family, friends, or the wider community might be an important resource for minority populations to preserve their health [31]. Proxies for social support, such as living in a relationship or being married, are associated with a lower risk of adverse cardiovascular events [32,33,34].

In the RLMS data, six categories were available for the variable level of education. The categories were *0–6 grades of comprehensive school; unfinished secondary education 7–8 grades of school]; unfinished secondary education [7–8 grades of school] plus something else, secondary school diploma; vocational secondary education diploma*; and *higher education diploma and more*. For descriptive statistics, the level of education was divided into three levels (non-completed secondary education, completed secondary education, and higher than secondary education) [35]. The employment status was measured by asking about the participants’ primary work at present. Categories of the RLMS data were *currently working; on paid leave [maternity leave or taking care of a child under 3 years of age]; on another kind of paid leave; on unpaid leave*; and *not working*. For descriptive analyses, employment was recoded into a variable with three categories (unemployed, paid or unpaid leave, and employed). *Never married; in a registered marriage; living together and not registered; divorced and not remarried; widower or widow; registered but not living together*; and *married* were the seven categories of the variable marital status. To describe frequency distributions of marital status, the seven categories were condensed into four (living alone, divorced or widowed, living together but not married, and living together and married). Smoking, diabetes, and subjective health were only considered mediators for the association of self-reported nationality with healthcare utilization. Subjective health was used as variable with the original five categories.

### 2.3. Statistical Methods

For descriptive analysis, numbers of observations and weighted proportions together with 95% confidence intervals (CIs) were calculated. The clustered sampling design and survey weights were considered when calculating proportions and the 95% CI. All further analyses are from multilevel logistic regressions with primary sampling units as random effects to account for the clustered sampling design. In addition, survey weights were introduced in all regression models. Table 1 shows that missing data were below 2% in all variables. Therefore, regression analyses were performed on a complete dataset of 40,915 observations. Smoking, diabetes, subjective health, and the utilization of health check-ups were dependent variables. Subjective health was analyzed as a trichotomous variable. The category ‘average’ was used as a reference, and two logistic regression models were fitted which modelled: (1) odds of good versus average subjective health; and (2) odds of bad versus average subjective health. We chose this strategy because our analysis software did not allow the introduction of survey weights in multinomial multilevel regression. A stepwise modelling technique was employed for all outcomes, gradually introducing confounders (model 1) and mediators (social mediators: model 2, health-related mediators: model 3) in the models. For all confounders and mediators, we used the original categorizations of the variables provided by the RLMS to maintain maximum variability. One regression coefficient for each survey year was used. Finally, we investigated effect measure modifications of the association of self-reported German nationality with the four outcomes by period of the survey (1994–2005 vs. 2006–2018). These time periods were chosen to gain a high power when assessing the change in odds ratios over time. Studying effect modification by survey year is important, because our study covered a very long period in which important changes in associations might have occurred. In association with the utilization of health check-ups, we additionally investigated effect measure modifications by place of residency, because a rural infrastructure may represent a substantial barrier to healthcare in Russia [36]. Subsequently, the odds of all defined outcomes among ethnic Germans were compared with the odds among non-Germans (referred to as “other” in all tables) as the reference group using odds ratios (ORs) and 95% CIs. P-values were calculated using Wald tests. Analyses were conducted in R, version 4.0.2, and multilevel regression models were calculated calling MLwiN, version 3.05, from within R [37,38,39].

### 2.4. Ethics Statement

The ethics committee of the Medical University of Brandenburg, Brandenburg an der Havel, Germany, approved the ethics application, with the number E-01-20191119.

## 3. Results

A total of 41,833 observations were analyzed, 158 of whom were ethnic Germans.

There was a higher proportion of ethnic Germans in the age groups 40–59, 60–79, and over 80, than the general population of Russia (Table 1). The proportion of males among ethnic Germans was higher than among non-Germans. A further difference was found in the place of residence of the respondents: 34.8% lived in small towns or villages. Among the non-Germans, this figure was 24.9%. Moreover, non-Germans had a higher level of education than ethnic Germans. Among non-Germans, 21.5% had an education classified as higher than secondary, whereas only 11.0% of ethnic Germans had this level of education. An unfinished secondary education was found among 35.8% of ethnic Germans and 21.9% of non-Germans. Among ethnic Germans, 46.0% were unemployed, compared with 41.7% among non-Germans. Finally, 11.7% of ethnic Germans were living alone compared with 20.8% among non-Germans.

### 3.1. Smoking

Briefly summarized, there were no noticeable difference between ethnic Germans and non-Germans concerning their smoking behavior (Table 1). After the multilevel logistic regression analysis, the unadjusted OR was 0.88 (CI 0.64–1.21) and hardly changed after adjusting for confounders as well as for social mediators (Table 2).

### 3.2. Diabetes

For self-reported diabetes, there was also no difference between both groups (Table 1). The crude OR equaled 1.12 (CI 0.56–2.25) and, after adjusting for confounding, decreased to 0.94 (CI 0.41–2.00) (Table 2). Subsequently adjusting for social mediators did not change the OR.

### 3.3. Subjective Health

Of the non-Germans, 2.9% reported to have very good health, and 35.7% reported to have good health (Table 1). In comparison, the figures for ethnic Germans were 2.0% for very good and 20.6% for good health. In the multilevel logistic regression analysis modeling odds for good (very good or good) versus average subjective health, the unadjusted OR for German ethnicity was 0.49 (95% CI 0.34–0.72; *p*-value < 0.001) (Table 2). The OR adjusted for confounders yielded 0.67 (95% CI 0.48–0.92; *p*-value 0.01). After additional adjustment for the social mediators (education, employment, and marital status), the OR was 0.68 (95% CI 0.49–0.93; *p*-value 0.02). Considering regression analyses modeling the odds for bad (very bad or bad) versus average subjective health, the unadjusted OR for German ethnicity was 0.97 (95% CI 0.63–1.47, *p*-value 0.87). The OR after stepwise adjustment for confounders and mediators did not change substantially.

### 3.4. Health Check-Ups

Ethnic Germans reportedly made less frequent use of health check-ups than non-Germans (Table 1). Following the multilevel logistic regression analysis, the unadjusted odds ratio was 0.76 (CI 0.48–1.19; *p*-value 0.23) (Table 2). After adjusting for confounders, the OR came to 0.84 (CI 0.54-1.30) and the *p*-value equaled 0.44. There were no meaningful changes in the OR following adjustment for social mediators (education, employment, and marital status) and health-related mediators (smoking, diabetes, and subjective health).

### 3.5. Stratified Analyses

In the years before 2006, the odds for smoking were 0.48 times less for Germans than non-Germans (Table 3). After 2006, the OR was 1.91 with a *p*-value <0.001 for interaction. Concerning diabetes, a *p*-value of 0.65 did not confirm a significant interaction of ethnicity and year of survey. Finally, the odds of good versus average subjective health among ethnic Germans were lower than in the general population of Russia in both investigated periods. When modeling the odds of good versus average subjective health, there was no evidence for an effect measure modification of the association of ethnicity by year of survey. There was also no indication of effect measure modification when modeling odds of bad versus average subjective health.

Finally, the results of stratified analyses for the outcome health check-up utilization are presented. In regard to the year of survey, both ORs were <1, and the *p*-value for interaction by year of survey was 0.95, providing no evidence for effect measure modification (Table 4). When regarding the place of residency, there was some evidence of an effect measure modification: in urban areas, there was no evidence for a difference between ethnic Germans and non-Germans participating in health check-ups; however, in rural areas, the odds for utilizing health check-ups among German were 57% lower compared with the general Russian population (Table 4).

## 4. Discussion

Thus far, there have hardly been any studies on the health of ethnic Germans in Russia. This study represents an initial attempt to gain a better understanding of this minority in comparison with the majority population and enables a cross-national perspective on the health of this unique migrant group of resettlers. Ethnic Germans in Russia, the population of origin of resettlers, were less likely to evaluate their health as good compared with the general population of Russia. There seems to be no difference between ethnic Germans in Russia and the general Russian population regarding diabetes and utilizing health check-ups. Moreover, lower odds of smoking among ethnic Germans compared with non-Germans in Russia before 2006 seems to reverse to higher odds after 2006. Additionally, in rural areas of Russia, there is some evidence that ethnic Germans might be less likely to participate in health check-ups than the general population.

Self-rated health is regarded as a meaningful predictor for morbidity, the future use of healthcare services, and mortality [8]. Social networks and further social determinants, such as educational attainment, are related to subjective health; however, these and further mediators did not explain differences in subjective health between ethnic Germans and the autochthonous general population of Russia in our study [35]. Hence, further factors that might shape health of minorities, such as socio-structural racism, need to be considered. Immigrants and minorities tend to be “othered”, and thus, face more exclusion from central social domains such as meaningful work or opportunities to increase household income. Social exclusion, in turn, can lead to barriers to healthcare and health-promoting resources [40]. At the same time the “othering” and excluding as such leads to more stress, fear, and experience of prejudice and violence, which might have a negative impact on health [11]. Even after adjusting for confounders and social mediators, ethnic Germans were less likely to evaluate their health as good compared with the general population of Russia. However, marital status, level of education, and employment status are only proxies, because we could not control for all possible socioeconomic influences and all forms of social support. Thus, the information in this case is limited. Several studies have found that immigrants in Germany rate their health worse than the autochthonous population: Rommel et al. found that even after adjusting for socioeconomic factors, women with a migration background rated their health significantly worse than women with no migration background [17]. Additionally, Ronellenfitsch and Razum showed that the effect of migration had the biggest effect on health satisfaction compared with other socioeconomic factors among Eastern European migrants, mainly resettlers [41]. These results coincide with studies in which resettlers from the FSU rated their health worse than the general German population [42]. This raises the question as to whether the subjective health of resettlers might have partly originated from their sending country and what role social exclusion and structural racism might play. Overall, regarding subjective health, migrants and ethnic minorities seem to be disadvantaged compared with the majority population after controlling for confounding, which is in line with the findings among migrants and ethnic minorities in Europe [12].

The smoking habits in ethnic Germans and the general population of Russia reversed after 2006. Considering that the overall level of smoking seems to have decreased worldwide, especially since the beginning of the 2000s [43,44], this is an unexpected finding. In Russia, smoking among men has decreased since the beginning of the 21st century, whereas smoking among women has increased, especially in older groups [45]. Overall, our findings suggest that smoking prevention among ethnic Germans in Russia should be carefully evaluated. Moreover, Reiss et al. described a higher smoking prevalence amongst male resettlers in Germany compared with the male German population, which adapts with increasing durations of stay, suggesting that smoking could be an imported behavior amongst resettlers [46]. In addition, lower smoking prevalence compared with the general Russian population before 2006 is in line with the finding of rather low cardiovascular mortality among resettlers in Germany [47]. Our finding would support the hypothesis that ethnic Germans are healthier compared with the Russian population at the time of migration to Germany, and that the high cardiovascular mortality which was expected among resettlers based on the large excess mortality in Russia in the 1990s was not observed [48].

Generally, there does not seem to be a difference in the utilization of health check-ups between Germans and non-Germans. In our results, however, place of residence seemed to be important. There was some evidence that Germans in rural areas participated less often in preventative examinations than non-Germans, whereas there was no difference in urban areas. However, the number of cases in rural areas was too low to yield a more precise estimate in this subgroup. Moreover, urbanized areas continue to benefit from the medical care infrastructure of the former Soviet Union [36]. At the same time, many medical facilities and small hospitals in rural areas were closed and a considerable number of beds were eliminated, which might have contributed to the observation that people in urbanized regions utilize preventative consultations by physicians more often than people in rural areas in Russia [36]. Furthermore, irregular access to public transport, prolonged traveling time, and lack of access to a car have frequently been described as a barrier to healthcare for ethnic minorities [49]. Finally, organized measures of secondary prevention have been available in Russia since the inclusion of periodic health check-ups (dispansertizatsiya) in the compulsory health insurance system in 2013 [50]. However, especially in rural areas with worse medical care [36], accessibility of the health check-up program might have been suboptimal for ethnic Germans.

### Limitations and Strengths

In this study, there was only a small sample of ethnic Germans. Thus, the power to identify differences between Germans and non-Germans was limited. Another limitation is the fact that all information provided by the participants is based on self-reported data. This leaves a possibility for systematic distortions such as social desirability bias. Moreover, we only covered a snapshot of risk factors for non-communicable diseases. It would be desirable to gain a more comprehensive picture of risk factors and disease burden in this population to detect health inequalities. Finally, more research is needed in additional countries of origin of resettlers. Russia is one of the most important countries of origin; however, data from Kazakhstan and Ukraine are needed to derive a more complete picture.

At the same time, this study offers first insights into the health of ethnic Germans in Russia, and thus provides further information about the population of origin of resettlers in Germany. The RLMS survey was designed to be nationally representative of Russia. We used survey weights to adjust for non-responses; hence, our results can be considered representative. The fact that ethnicity was self-reported is an additional strength of the study. Self-reported ethnicity might give a more comprehensive picture of how persons identify themselves rather than asking for formal citizenship. In addition, the participants were interviewed face-to-face, which is expected to yield high data quality.

## 5. Conclusions

This study provides the first results on the health status and healthcare utilization among ethnic Germans in Russia compared with the autochthonous population. A higher smoking prevalence between 2006 and 2018 suggests that primary prevention strategies specifically targeting smoking are important among this minority in Russia. Further studies with oversampling of ethnic Germans could aid in further identifying specific prevention needs. In addition, the lower use of health check-ups among ethnic Germans in rural regions highlights the importance of improving the accessibility of healthcare for this minority, especially in sparsely populated areas.

Finally, our study facilitates comparisons of health between resettlers in Germany and parts of their population of origin in Russia. Subjective health seems to be low, both in resettlers in Germany and among their population of origin in Russia. This finding might highlight the importance of a better social inclusion of minority populations in order to alleviate psychosocial stressors. Finally, cross-national research should be extended to additional countries of origin of resettlers such as Kazakhstan or Ukraine.

## Figures and Tables

**Table 1 ijerph-19-00166-t001:** Socio-demographic and health-related characteristics of the study population.

		Other (*n* = 41,675)		German (*n* = 158)	
		*n*	Prop. (95% CI)	*n*	Prop. (95% CI)
Sex	Female	23,080	52.9 (52.4–53.5)	77	46.5 (39.9–53.3)
	Male	18,595	47.1 (46.5–47.6)	81	53.5 (46.7–60.1)
Age	0–19	4112	11.0 (10.0–12.0)	12	8.79 (4.4–16.6)
	20–39	18,485	46.2 (44.9–47.5)	51	34.6 (26.8–43.3)
	40–59	11,630	27.9 (26.9–28.8)	54	35.6 (27.4–44.8)
	60–79	6437	12.9 (12.1–13.8)	36	18.5 (13.1–25.6)
	80+	1005	2.0 (1.8–2.3)	5	2.5 (0.8–7.6)
	Missing	6		0	
Place of residency	Urban	31,907	77.9 (64.7–87.1)	103	66.7 (39.2–86.2)
	Rural	9768	22.1 (12.9–35.3)	55	33.3 (13.8–60.8)
Year of survey	1994–1999	12,352	30.8 (27.5–34.2)	72	49.3 (40.7–57.9)
	2000–2005	6373	13.4 (9.7–18.3)	27	15.9 (9.3–25.7)
	2006–2011	14,788	36.0 (33.4–38.7)	36	21.1 (15.5–28.2)
	2012–2018	8162	19.8 (17.8–22.0)	23	13.7 (8.5–21.4)
Level of education	Unfinished secondary	9335	21.9 (19.3–24.8)	55	35.8 (26.2–46.7)
	Completed secondary	23,420	56.6 (55.1–58.0)	85	53.2 (44.8–61.5)
	Higher than secondary	8808	21.5 (18.4–25.0)	15	11.0 (6.7–17.6)
	Missing	112		3	
Employment	Unemployed	17,892	41.7 (39.9–43.4)	75	46.0(35.5–56.9)
	Paid or unpaid leave	1273	3.1 (2.8–3.4)	3	2.2 (0.8–5.6)
	Employed	22,465	55.3 (53.5–57.0)	80	51.8 (40.6–62.9)
	Missing	45		0	
Marital status	Living alone	7581	20.8 (19.2–22.4)	17	11.7 (6.8–19.4)
	Divorced or widowed	7381	16.5 (15.7–17.4)	36	21.8 (13.5–33.3)
	Living together, not married	5178	11.2 (10.3–12.2)	24	13.5 (9.1–19.6)
	Living together, married	21,112	51.5 (49.4–53.5)	80	53.0 (42.5–63.3)
	Missing	423		1	
Smoking	No	27,142	64.2 (62.4–66.0)	104	66.1 (57.4–73.7)
	Yes	14,498	35.8 (34.0–37.6)	54	33.9 (26.3–42.6)
	Missing	423		0	
Diabetes	No	39,779	96.1 (95.6–96.5)	149	96.4 (91.5–98.5)
	Yes	1773	3.9 (3.5–4.4)	7	3.6 (1.5–8.5)
	Missing	123		2	
Subjective health	Very good	1105	2.9 (2.3–3.7)	3	2.0 (0.7–5.9)
	Good	14,189	35.7 (34.1–37.4)	30	20.6 (14.2–29.0)
	Average	20,747	49.3 (47.4–51.2)	99	62.4 (53.7–70.4)
	Bad	4709	10.5 (9.9–11.0)	19	11.8 (7.1–19.0)
	Very bad	744	1.6 (1.4–1.9)	7	3.1 (1.4–6.6)
	Missing	181		0	
Health check-up	No	33,482	80.3 (79.0–81.6)	132	83.1 (74.3–89.3)
	Yes	8147	19.7 (18.4–21.0)	26	16.9 (10.7–25.7)
	Missing	46		0	

Prop: weighted proportions (in %) using survey weights of the RLMS. 95% CI: 95% confidence interval.

**Table 2 ijerph-19-00166-t002:** Crude and adjusted associations of self-reported ethnicity with smoking, diabetes, subjective health, and the utilization of health check-ups.

					MODEL 1			MODEL 2			MODEL 3		
		Crude OR	95% CI	*p*-Value	Adjusted OR	95% Cl	*p*-Value	Adjusted OR	95% CI	*p*-Value	Adjusted OR	95% CI	*p*-Value
Smoking	Other	1.00 (ref.)			1.00 (ref.)			1.00 (ref.)					
	German	0.88	0.64–1.21	0.43	0.82	0.55–1.21	0.31	0.75	0.53–1.08	0.12			
Diabetes	Other	1.00 (ref.)			1.00 (ref.)			1.00 (ref.)					
	German	1.12	0.56–2.25	0.74	0.94	0.41–2.00	0.89	0.94	0.39–2.25	0.89			
Subjective Health	(1) Models for odds of good vs. average subjective health (*n* = 35,537)												
	Other	1.00 (ref.)			1.00 (ref.)			1.00 (ref.)					
	German	0.49	0.34–0.72	<0.001	0.67	0.48–0.92	0.01	0.68	0.49–0.93	0.02			
	(2) Models for odds of bad vs. average subjective health (*n* = 25,880)												
	Other	1.00 (ref.)			1.00 (ref.)			1.00 (ref.)					
	German	0.97	0.63–1.47	0.87	0.77	0.47–1.25	0.29	0.69	0.42–1.14	0.15			
Health check-up	Other	1.00 (ref.)			1.00 (ref.)			1.00 (ref.)			1.00 (ref.)		
	German	0.76	0.48–1.19	0.23	0.84	0.54–1.30	0.44	0.86	0.56–1.31	0.48	0.85	0.55–1.31	0.46

Model 1: Odds ratio adjusted for year of survey, sex, place of residency, and age (confounders). Model 2: Odds ratio adjusted for year of survey, sex, place of residency, age, education, employment, and marital status (confounders and social mediators). Model 3: Odds ratio adjusted for year of survey, sex, place of residency, age, education, employment, marital status, smoking, diabetes, and subjective health (confounders, social mediators, and health mediators). OR, odds ratio. CI, 95% confidence interval.

**Table 3 ijerph-19-00166-t003:** Effect modification of the association of self-reported ethnicity with smoking, having diabetes, and subjective health by year of survey.

	Year of Survey	OR	95% CI	*p*-Value
Smoking	<2006			
	Other	1.00 (ref.)		
	German	0.48	0.28–0.82	
	≥2006			
	Other	1.00 (ref.)		
	German	1.91	1.09–3.37	<0.001
Diabetes	<2006			
	Other	1.00 (ref.)		
	German	1.18	0.44–3.20	
	≥2006			
	Other	1 (ref.)		
	German	0.73	0.15–3.61	0.65
Subjective Health	(1) Model for odds of good vs. average subjective health (*n* = 35,537)			
	<2006			
	Other	1.00 (ref.)		
	German	0.71	0.45–1.12	
	≥2006			
	Other	1.00 (ref.)		
	German	0.51	0.30–0.86	0.39
	(2) Model for odds of bad vs. average subjective health (*n* = 25,880)			
	<2006			
	Other	1.00 (ref.)		
	German	0.66	0.36–1.20	
	≥2006			
	Other	1.00 (ref.)		
	German	1.03	0.41–2.59	0.46

OR, odds ratio. *p*-value: *p*-value for interaction. CI, 95% confidence interval. All models have been adjusted for confounding: sex, place of residency, and age.

**Table 4 ijerph-19-00166-t004:** Effect modification of the association of self-reported ethnicity with the utilization of health check-ups by year of survey and place of residency.

	Health Check-Up	OR	95% CI	*p*-Value
Year of survey ^1^	<2006			
	Other	1.00 (ref.)		
	German	0.86	0.42–1.48	
	≥2006			
	Not German	1.00 (ref.)		
	German	0.83	0.38–1.82	0.95
Place of residency ^2^	Urban			
	Other	1.00 (ref.)		
	German	1.03	0.65–1.66	
	Rural			
	Other	1.00 (ref.)		
	German	0.43	0.21–0.87	0.03

OR, odds ratio. *p*-value: *p*-value for interaction. CI, 95% confidence interval. ^1^ Model adjusted for confounding: sex, place of residency, and age. ^2^ Model adjusted for confounding: sex, year of survey, and age.

## Data Availability

The data presented in this study are openly available in the Russia Longitudinal Monitoring Survey Dataverse (RLMS-HSE Longitudinal Data Files) at https://doi.org/10.15139/S3/12438.

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
