# Peer review of "The Health Status and Healthcare Utilization of Ethnic Germans in Russia"

_ijerph, 2021, doi:10.3390/ijerph19010166_

Round 1

Reviewer 1 Report

Thank you for the opportunity to review this manuscript. I enjoyed reading the paper and I think the paper studies an important topic – the health of ethnic Germans in Russia. Overall, I think the paper is well written, it contributes to our collective knowledge about public health of ethnic minorities, and the main results are interesting.

However, for me, much of the research design would benefit from further elaboration and justification for many of the decisions made for the analysis. There are three main issues I would recommend addressing in future iterations of the paper.

First, several variables are dichotomized and it isn’t clear why this is the best decision. For instance, the likert scale for subjective health is dichotomized into good versus average – I think this complicates the interpretation of the results more than it illuminates them, especially as ‘average’ responses are combined with ‘bad’ and ‘very bad’ responses. I would recommend maintaining the five different responses in the analysis, but at the very least having a trichotomous variable would increase measurement validity, especially given the modal response is ‘average’ in the sample. Similarly, variation for other variables is reduced for recoded variables such as education without any theoretical justification. I would recommend either or both of two courses of action: 1) maintaining variation in the variables and running analysis using those, perhaps reporting results in the appendix, and 2) making a theoretically informed argument in the paper about why these are the correct decisions for the analysis you’re conducting.

Second, on a similar note, it isn’t clear why the survey periods are dichotomized into pre- and post-2006. Did a significant event or piece of legislation occur then that would make this comparison meaningful? If not, I would recommend conducting cross-sectional analysis across the entire period and avoid subsetting the data in the absence of a good reason for doing this.

Third, as the authors note, the number of observations for ethnic Germans in the sample is tiny (158 or an average of about 6.5 per survey year). This means that comparisons are pretty fraught without introducing survey weights at the very least, and it isn’t clear from the analysis why the authors did not include post-stratification to at least adjust for this. Again, some justification for this would be helpful, but I would prefer that survey weights and/or post-stratification is used to make the reader more confident about the results of the analysis.

Finally, much of the framing of the paper highlights the relative poorer health among Resettlers in Germany, but the paper’s data and analysis doesn’t speak to that directly. I would recommend focusing in the introduction and the conclusions on what the data does address – namely the health of ethnic Germans in Russia – which is itself worthy of further analysis and understanding rather than highlighting the contribution as understanding the relative disparities in Germany which cannot be addressed using the data presented in this paper.

Overall, this is a good paper that I enjoyed reading, and I hope the authors are able to make these changes to the manuscript to improve it further.

Reviewer 2 Report

This paper aims to use data from the Russian Longitudinal Monitoring Survey II to describe the health status and health care utilization of ethnic Germans in Russia.  Authors assert a need for this study due to the fact that ethnic German settlers from the former Soviet Union compose one of the largest ethnic minority groups in Germany, though show less favorable health outcomes and utilization of preventive services compared to the majority Russian population.  Authors provide a rationale for their study by stating there is a lack of studies on the health of ethnic Germans in Russia. Feedback is as follows:

  1. The word ‘Resettlers’ is capitalized throughout the paper. Is the word meant to be presented as a pronoun?  The authors should clarify.
  2. Authors present a compelling introduction with the discussion of the global burden of noncommunicable disease, need to explore social determinants of health as risk factors, and importance of assessing the impact of migration on health.
  3. Line 70 – Check spacing in the word “Theyrepresented”.
  4. Line 76 – The word ‘autochthonous’ should be defined for the lay reader.
  5. The methodology is well-described, and the statistical procedures used are sound.
  6. For the Conclusions, future research may also consider exploring and stratifying outcomes by the participant’s country of origin, since authors note that resettlers come from Eastern European, Northern and Central Asian countries (line 66).
  7. Overall, this is a pertinent, insightful, and unique study. It is relevant and interesting to read.  The background and significance of the study is well-established.  The research is well conducted.

Round 2

Reviewer 1 Report

Thank you for the opportunity to review a revised version of this manuscript. I appreciate the efforts the authors made to address the recommendations made in the previous round of review, and I recommend the paper for publication - I think it makes an interesting and important contribution to our understanding of public health among ethnic Germans in Russia.

This manuscript is a resubmission of an earlier submission. The following is a list of the peer review reports and author responses from that submission.

Round 1

Reviewer 1 Report

This manuscript presents the result of investigation on difference between ethnic Germans living in Russia and the Russian population in terms of their health status such as smoking behavior, diabetes, and subjective health as well as their utilization of health check-ups. The topic is interesting and the investigation is also useful for health policy development to promote a good health status among Germans living in Russia. However, there are some concerns which should be addressed before the acceptance of publication in this journal. I have following comments.

  1. In the abstract, the number of cases or observations must be indicated.
  2. I am curious why the analysis was divided into two periods such as 1994-2005 and 2006-2018 ? please explain a concrete reason in the introduction.
  3. It is important to provide explanations on why smoking behavior, diabetes, subjective health, and utilization of health check-ups were investigated. Besides of available data, authors should provide more explanations stating significance of investigations on these health status.
  4. The proposed conceptual framework is not supported by theoretical discussion. Please discuss how those mediators and confounders could influence all dependence variables (such as smoking behavior, diabetes, subjective health, and utilization of health check-ups) by reviewing relevant literature and making discussions.
  5. I am concerned about missing values in each variable which are quite big. For instance the missing values of smoking behavior among non German group is counted for 23.9%.  Missing values of health check-up among non German group is counted for 2.1%. Please explain how these missing values were managed. All raw data need to be inspected and checked before the analysis. In cause that an observation has so many values, it must be excluded from the analysis.
  6. Page 3, it is stated that the reported nationalities “German” and “German-Jew” were considered as German nationality. All other nationalities including no reported nationality were contained in the comparison group. I am concerned that all nationalities without “German” and  “German-Jew” were combined together to represent general Russian population. This could make the analysis unconvincing and bias. Please eliminate observations with no reported nationality and other non-Russian cases from the analysis. Each nationality has its own culture and health related habits.
  7. Regarding the table 2, though the study focuses on ethicality (German and non-German), OR, CI, and P-value of all independent variables should be reported in Model 1, 2, and 3.  It can be reported as supplementary data/appendix or in the text itself.
  8. In table 4, I could not see the P-value of 0.51, but it is stated in the text “The p-value of 0.51 provided little evidence for effect measure modification (Table 4)”. Please report the results in accordance to the statistic results shown in the table. If the analysis shown no significant effect, that means no effect.

Reviewer 2 Report

Please calmly consider whether the representativeness of your sample is adequate to extrapolate the results to the entire country, 63 PSU out of 98 in just three metropolitan areas could be a limitation that should not be overlooked.